# Mineral Reaction Kinetics during Aciding of the Gaoyuzhuang Carbonate Geothermal Reservoir in the Xiong'an New Area, Northern China

Gaofan Yue [1,2] , Xi Zhu [1,2,*], Guiling Wang [1,2] and Feng Ma [1,2]

1   Institute of Hydrogeology and Environmental Geology, Chinese Academy of Geological Sciences, Shijiazhuang 050061, China
2   Technology Innovation Center of Geothermal & Hot Dry Rock Exploration and Development, Ministry of Natural Resources, Shijiazhuang 050061, China
*   Correspondence: zx19860727@163.com

**Abstract:** There are abundant geothermal resources in the Xiong'an New Area, China. Drilling has revealed a greater potential in the deep Gaoyuzhuang geothermal reservoir. However, the reservoir required acidification to increase its water production. In this study, three types of core samples with different mineral compositions from different depths in the target boreholes were selected for acid rock reaction experiments at the temperature of 40 °C, 60 °C, 80 °C and 100 °C, and pressure of 30 MPa. The kinetics of the acid rock reaction of the major minerals were modeled based on the transitional state theory. The kinetic parameters were obtained by comparing the modelling and experimental results. The results show that the lithology of the Gaoyuzhuang reservoir is primarily dolomite. The dissolution ratio for 15 wt.% HCl reached 84.1% on average for the rock fragments. Temperature has a significant effect on the dissolution rate of the minerals. In the presence of HCl (acidic mechanism), the reaction rate constants of the dolomite, calcite and illite reached $2.4 \times 10^{-4} \text{ mol/m}^2/\text{s}$, $5.3 \times 10^{-1} \text{ mol/m}^2/\text{s}$ and $9.5 \times 10^{-2} \text{ mol/m}^2/\text{s}$, respectively. The results of this study provide the basic parameters for the design and evaluation of field acidizing.

**Keywords:** carbonate geothermal reservoir; acid-rock reaction kinetics; reaction vessel experiment; numerical simulation

## 1. Introduction

Geothermal energy will be one of the most important energy resources in the future. There are abundant geothermal resources in the Xiong'an New Area, China. The previously developed and utilized reservoirs include the Neoproterozoic Minghuazhen Formation (Nm), the Neoproterozoic Guantao Formation (Ng), the Paleoproterozoic Dongying Formation (E), the Mesozoic Cambrian–Ordovician strata (O-∈), and the Jixian System Wumishan Formation (Jxw) [1–5]. Geothermal boreholes implemented by the China Geological Survey in the Xiong'an New Area have encountered the Gaoyuzhuang reservoir in the Jixian System. Research has shown that the Gaoyuzhuang geothermal reservoir has a higher temperature and larger geothermal capacity than the other reservoirs [6,7]. Moreover, its development and utilization will not cause geological and environmental problems [8–10]. Due to the influence of the regional tectonics and natural karstification, the fracture distribution, type, opening width and interconnection in the Gaoyuzhuang Formation are not homogeneous, leading to non-homogeneous permeability [11]. The water production from some wells cannot meet the demand of large-scale development, and thus reservoir modification is necessary.

Acidizing or acid fracturing is one of the most common and economically effective methods of increasing the productivity of carbonate reservoirs. The transformation of carbonate geothermal reservoirs in northern China began with the Cambrian and Ordovician reservoirs in Beijing [12] and Tianjin [13]. With the increase in development depth,

this technology has also been adopted in the Jixian System in the Wushan Formation. For example, in geothermal well DL-24 in Tianjin, the maximum water production was increased from 100 m$^3$/h to 157 m$^3$/h and the water temperature was increased from 81 °C to 89 °C after acidizing [14].

The principle of acidification is to use hydrochloric acid to react with carbonate minerals (e.g., dolomite and calcite) in order to remove natural fracture blockages and expand natural fracture apertures, creating a highly permeable area around the geothermal well and increasing the water discharge. The dissolution rate of dolomite depends on the characteristics of the rock, such as the mineral composition, degree of crystallinity and specific surface area [15]. The dissolution rate of dolomite tends to be lower than that of calcite [16–18] due to the presence of magnesium [19–21]. However, higher magnesium levels do not always lead to lower reaction rates [22]. Plummer [23] found that the reaction rate of dolomite depends on the reaction of H$^+$, H$_2$O and H$_2$CO$_3$ with the rock surfaces. Gautelier [19] observed that when the pH is less than four, the reaction rate is mainly determined by the reaction of H$^+$ with the rock surface. For calcite and dolomite, the dissolution rates are proportional to and fractional powers of the rock surface H$^+$ activity, respectively. The effect of the pH on the reaction rate varies from case to case. At pH < 1, the dissolution rates are limited by surface reactions. As the pH increases, the role of diffusive transport becomes increasingly important, suggesting H$^+$ saturation of the mineral surfaces occurs at low pH values [24]. Guo [25] conducted chemical kinetics experiments using single calcite, dolomite and carbonic acid at different temperatures and pH values. The results showed that for Ca$^{2+}$, the amount of calcite dissolved was larger than that of dolomite. For dolomite, the dissolution of Mg$^{2+}$ was lower than that of Ca$^{2+}$. The chemical kinetics model developed revealed that the higher the temperature is, the larger the reaction orders are, and the greater the effect of the concentration on the chemical reaction is. Conversely, the smaller the pH, the smaller the activation energy of the reaction and the greater the reaction rate. Although many studies have been conducted on the mechanism of acid–rock reactions [26,27], the kinetic performance of acid–rock reaction under specific geological conditions (temperature, pressure and mineral composition) varies significantly, which directly affects the reservoir modification.

In this study, a combination of experiments, numerical simulations and data analysis was used to analyze the mineral dissolution process and its kinetic characteristics during the acidification of geothermal reservoir rocks from the Gaoyuzhuang Formation, and then, the main influencing factors were investigated. First, core samples with different mineral compositions were selected for in situ high-temperature acid–rock reaction experiments to obtain the key ion concentrations of the residual acid after different reaction times and to clarify the reaction process. Second, based on transition state theory, acid–rock reaction kinetic models were developed to simulate the mineral dissolution process. It was found that the simulation and experimental results corroborate each other and reveal the mechanism of the acid–rock reaction of the Gaoyuzhuang Formation. Finally, the kinetic parameters of the acid rock reaction corresponding to the main minerals were derived. The research results of this study provide support for field acidizing tests and numerical simulations.

## 2. Geological Conditions

The Xiong'an New Area is located in the northern part of the Jizhong Depression (Level III) within the North China Basin (Level II) on the China-Dynasty Quasi-Terrane (Level I) (Figure 1a,b) [28]. According to previous studies, the Bohai Bay Basin has experienced a series of tectonic movements since the Paleocene, the most influential among which are the Qingyu, Jixian, Caledonian, Indo-Chinese, Yanshan and Himalayan movements [29–31]. These tectonic movements led to stratigraphic uplift and denudation. The average Moho surface depth in the Xiong'an New Area is only 33 km, which is highly advantageous for the conduction of mantle source heat to the shallow strata [2,32,33].

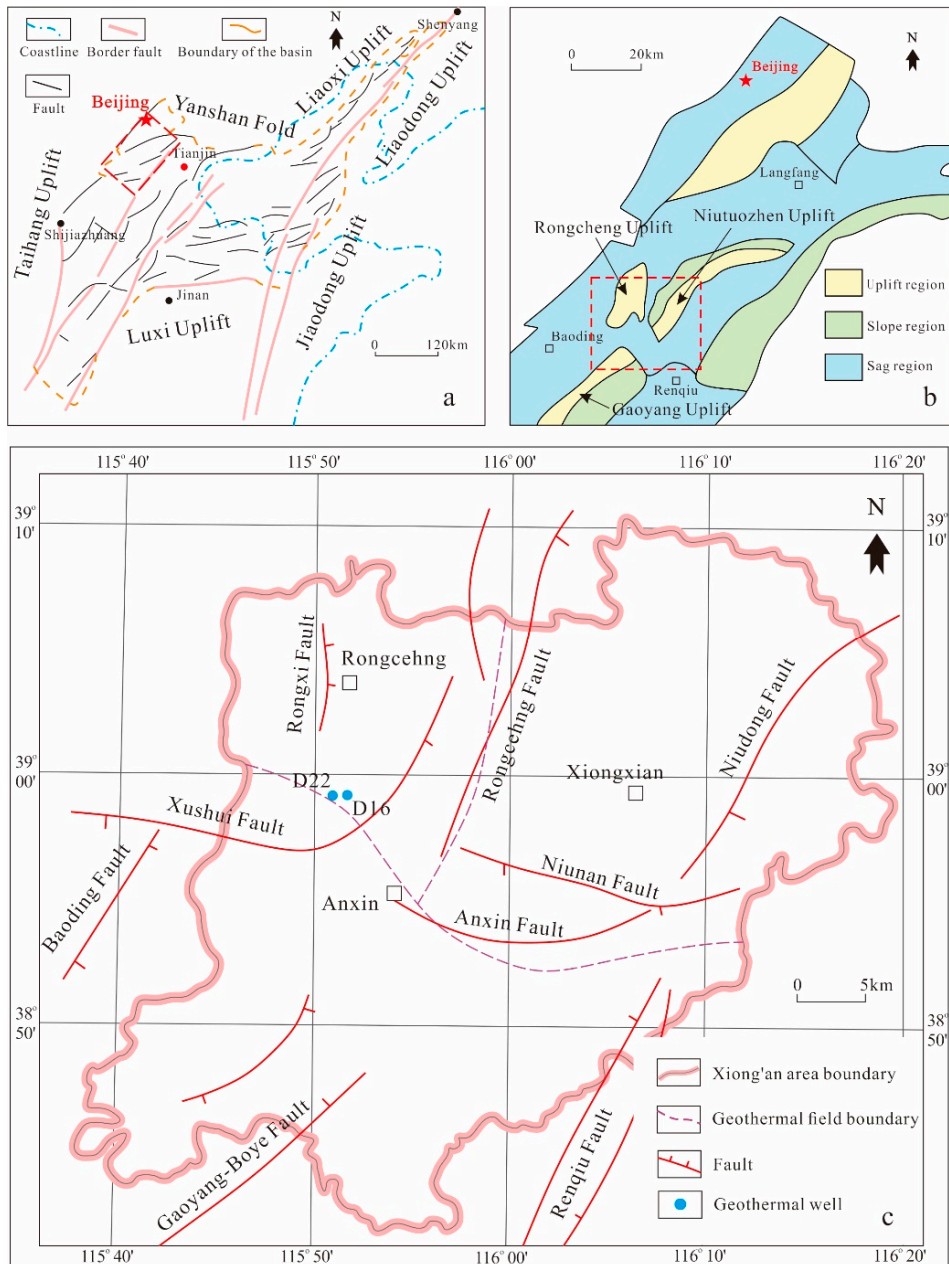

**Figure 1.** (**a**) Basic structural map of the Bohai Bay Basin; (**b**) Basic structural map of the Jizhong Depression [28]; and (**c**) Map showing the distributions of the major faults, geothermal fields, and boreholes in the Xiong'an New Area.

The bedrock undulations demonstrate that the secondary tectonic units are well developed, such as the Niutuozhen bulge, the Rongcheng bulge, and the Gaoyang bulge [31,34]. The faults are well developed, such as the Rongdong fault, Xushui fault, Niudong fault, Niunan fault and Gaoyang fault [1]. These faults control the regional tectonic pattern and are of great significance to the formation of the deep geothermal system in the Xiong'an New Area. Our research area is located in the southern part of the Rongcheng bulge.

The Gaoyuzhuang Formation belongs to the lower section of the Jixian system and is distributed throughout the Rongcheng geothermal field (Figure 1c). Well D22 in the study area intersects the Gaoyuzhuang Formation.

## 3. Methodology

### 3.1. Materials

The rock used in the experiment was gray–white dolomite, which was collected from well D22 in the Gaoyuzhuang Formation reservoir in the Xiong'an New Area (Figure 2). The collection depth was 2000–3500 m.

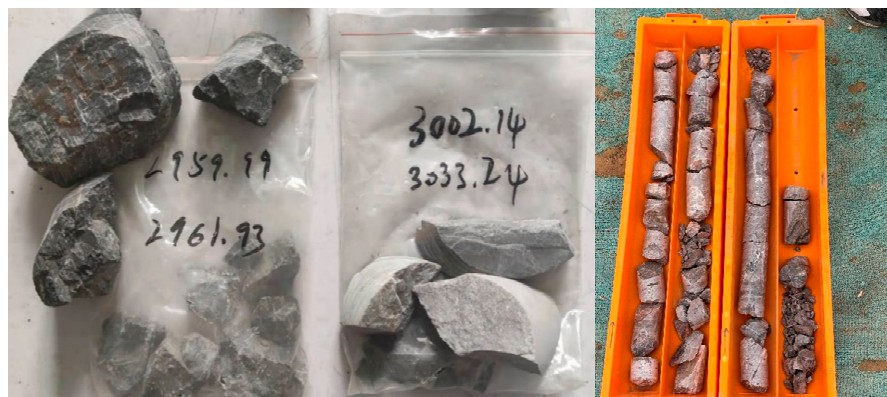

**Figure 2.** Core samples of the Gaoyuzhuang Formation from wells D22 and D16 in the Xiongan New Area.

The mineral compositions of the core samples were obtained via rock-thin section identification and whole rock X-ray diffraction (XRD) analysis. The typical rock types of the Gaoyuzhuang Formation in well D22 mainly include four types. (1) Powder dolomite with dolomite as the main mineral: The dolomite content is greater than 99%, and the fractures filled with carbonate minerals in the later stage can be seen. (2) Siliceous powdery dolomite: The siliceous content is higher—up to 70%—the dolomite content is lower, and the content of the other types of minerals is still low. (3) Clayey silty micrite dolomite: Dolomite is still the main mineral. The clay mineral content is higher, generally up to 15–20%, and the clay minerals are mainly illite. (4) Clayey sandy argillaceous dolomitic limestone: It is mainly characterized by increased calcite (up to 20%) and terrigenous sand debris contents. The terrigenous sand debris is mainly composed of microcline and plagioclase, with clay minerals accounting for 15%. Table 1 presents the specific identification results.

**Table 1.** Identification of rock ore in the Xiong'an New Area.

| Serial Number | Lithological Name | Main Components | Picture |
|:---:|:---:|:---:|:---:|
| 1 | Silty dolomite | Dolomite (>99%): semi-autohedral rhombohedrons arranged in a mosaic shape, form the main body of the rock. The fractures are filled by late carbonate minerals, which are also found in the rock. | |
| 2 | Laminated siliceous silty dolomite | It is composed of dolomite (45%) and siliceous material (55%), and the two components are alternately distributed in strips and stripes, forming horizontal to microwave laminations with different widths, i.e., a lamination structure. The laminae are sometimes interspersed with each other. | |

**Table 1.** *Cont.*

| Serial Number | Lithological Name | Main Components | Picture |
|---|---|---|---|
| 3 | Argillaceous silty micrite dolomite | Dolomite (75–80%): it is mainly allomorphic granular, and the aggregates are distributed in strips and laminations. It is the main body of the rock. Clayey soil (15–20%): it is composed of cryptocrystalline micro-scale clayey minerals, which are enriched and distributed in strips. There are cracks filled with dolomite and gypsum in the rock. |  |
| 4 | Clayey sandy argillaceous dolomite | The dolomite (40%) is hemihedral rhombohedral allomorphic granular in form; Calcite (20%): it is hemihedral rhombohedron—allomorphic granular, with limonitization in the later stage. Terrigenous sand debris (25%): it is composed of quartz, potassium feldspar and plagioclase. The type of potassium feldspar is microcline, with kaolinization and limonitization in the later stage. Clayey soil (15%). |  |

It can be seen from the X-ray diffraction analysis results of the core samples (Table 2 and Figure 3) that the main mineral of the thermal reservoir rocks of the Gaoyuzhuang Formation is dolomite, followed by quartz, and a small amount of clay minerals or feldspar and calcite. These results are similar to those of the thin section identification.

**Table 2.** Whole rock X-ray Diffraction analysis results for the core samples from the Xiong'an New Area.

| Sample Number | Sample Depth | Dolomite (%) | Quartz (%) | Clay (%) | K-Feldspar (%) | Plagioclase (%) | Calcite (%) |
|---|---|---|---|---|---|---|---|
| R1 | 2904–2906 | 70 | 5 | 10 | 5 | 5 | 5 |
| R2 | 3048–3050 | 99 | 0 | 0 | | 0 | |
| R3 | 3050–3052 | 80 | 14 | 5 | | 1 | |
| R4 | 3108–3110 | 75 | 15 | 6 | | 1 | |

*3.2. Experimental Strategy*

The temperature in well D22 in the Gaoyuzhuang Formation is about 60 °C at 3000 m depth. At present, the HCl concentrations used for carbonate reservoir acidizing are mainly 15 wt.% and 20 wt.%. In order to conserve valuable core resources, the rock debris dissolution ratio experiment was carried out at 60 °C to determine the reasonable acid concentration. Then, the high-temperature reactor experiment was carried out. The temperature range of the reactor experiment was appropriately expanded to cover the temperature range of the entire geothermal reservoir, including 40 °C, 60 °C, 80 °C and 100 °C. Since the mineral compositions of samples R3 and R4 were similar, only samples R1, R2 and R3 were selected for the experiments. The experimental scheme is presented in Table 3.

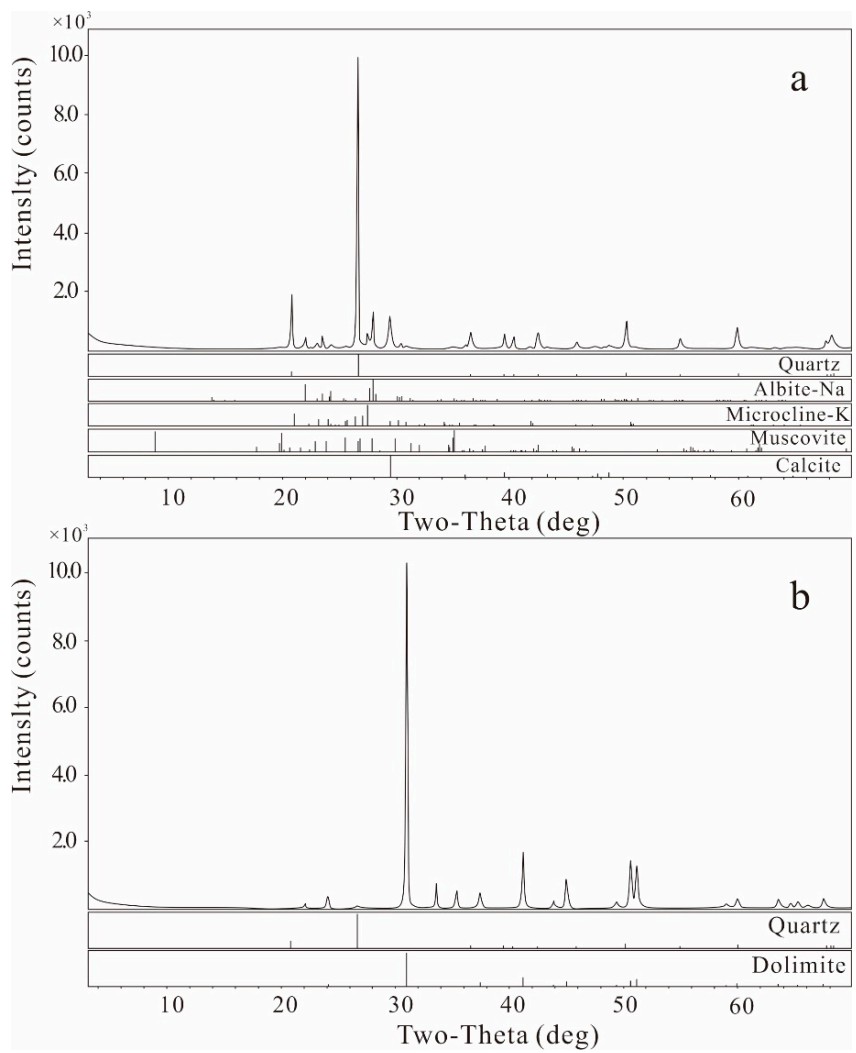

**Figure 3.** Energy spectrum analysis chart corresponding to cores (**a**) R1 and (**b**) R2.

**Table 3.** Experimental scheme of reactor.

| Serial Number | Core Number | Reaction Temperature °C |
|---|---|---|
| S01 | | 40 |
| S02 | R1 | 60 |
| S03 | | 80 |
| S04 | | 100 |
| S05 | | 40 |
| S06 | R2 | 60 |
| S07 | | 80 |
| S08 | | 100 |
| S09 | | 40 |
| S10 | R3 | 60 |
| S11 | | 80 |
| S12 | | 100 |

### 3.2.1. Rock Debris Dissolution Ratio

The rock debris was crushed, fully mixed and screened (80 mesh). About 25 g of sample were weighed, and 500 mL of HCl solution of different concentrations (15 wt.% and 20 wt.%) were added. The mixture was preheated to 60 °C and was let stand for 60 min. The filter equipment was prepared, and the filter paper was weighed. The mixture of acid

and sample was filtered and rinsed with deionized water. The filtered sample was dried and weighed. The dissolution ratio was calculated based on the mass difference:

$$\text{Dissolution ratio\%} = \frac{\text{Initial quality} - \text{Final quality}}{\text{Initial quality}} \times 100 \qquad (1)$$

### 3.2.2. Reactor Experiment

Different core samples were crushed and screened (5 mesh). HCl solution was added to the reaction vessel and preheated to the reaction temperature. Nitrogen was filled into the reactor to 30 MPa, representing the reservoir pressure. Around 25 g of the sample was weighed and added to the reaction vessel, which was then quickly sealed and pressurized. For the of 40 °C and 60 °C conditions, 10 mL of acid solution was collected at 2 min, 5 min, 10 min, 15 min, 20 min, 30 min, 45 min and 60 min. A 20-milliliter syringe was used to pass the acid through a 2-micrometer sieve head and place it in a pollution-free sampling bottle. For the 80 °C and 100 °C conditions, the sampling times were 1 min, 3 min, 5 min, 7 min, 10 min, 15 min, 20 min, 30 min, 45 min and 60 min.

A high-temperature and high-pressure reaction vessel produced by century Senlang was used as the experimental instrument. The inner tank of the vessel is forged using Hastelloy alloy to prevent acid–base corrosion. The reaction vessel is heated using a heating collar, the maximum temperature can reach 300 °C, and the temperature control accuracy is 0.1 °C. The maximum pressure is 60 MPa. The maximum volume is 500 mL. A magnetic stirring rod is provided to realize full contact and mixing of the reactants. There is one air inlet and one sampling interface. The structure of the reactor is shown in Figure 4.

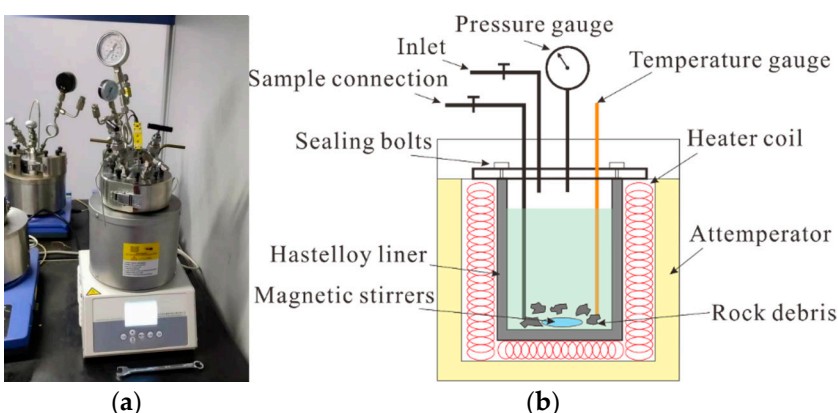

(**a**) (**b**)

**Figure 4.** (**a**) Equipment and (**b**) structure of the reactor.

### 3.2.3. Ion Concentration Analysis

The main minerals in the rocks of the Gaoyuzhuang Formation are dolomite, quartz, calcite, feldspar and clay minerals, hence the main ions after reaction were $Ca^{2+}$, $Mg^{2+}$, $Al^{3+}$, $H_4SiO_4$ and $HCO_3^-$. The $HCO_3^-$ was controlled by the carbonic acid equilibrium ($CO_2$ partial pressure), and the solution $HCO_3^-$ after the depressurization was unable to truly represent the concentration under formation conditions. Therefore, in this study, the concentrations of $Ca^{2+}$, $Mg^{2+}$, $Al^{3+}$ and $H_4SiO_4$ were mainly tested after the reaction. The testing unit was produced by Stande Testing Group Co., Ltd. (Qingdao, China), and the testing basis was the standard (GB/T 30902–2014) for the determination of impurity elements in inorganic chemical products via inductively coupled plasma emission spectrometry (ICP-OES).

### 3.3. Mineral Reaction Kinetic Model

The kinetic models used to describe the reaction rate of the minerals included the transition state theoretical model [35], the model based on activated surface complexes [36], the step wave model [37], and the nucleation theoretical model [38]. Among them, the

transition state theory (TST) model is the most widely used model. Other models are often only suitable for very simple cases (such as single mineral reactions) due to their complex forms and high computational costs.

### 3.3.1. Mineral Reaction Rate

Based on transition state theory, the general form of the reaction rate of the minerals can be expressed as follows [39]:

$$r_n = \pm k_n S_n \left| 1 - \Omega_n^\theta \right|^\eta \tag{2}$$

where $r_n$ $\left( \text{mol s}^{-1} \text{ kg w}^{-1} \right)$ is the mineral reaction rate. A positive value indicates dissolution of mineral N and a negative value indicates precipitation. $k_n$ $\left( \text{mol m}^{-2} \text{ s}^{-1} \right)$ is the kinetic dissolution or precipitation rate constant. $S_n$ $\left( \text{m}^2 \text{ kg w}^{-1} \right)$ is the reaction surface area. $\Omega_n$ is the mineral saturation value. $\theta$ and $\eta$ are empirical values describing the relationship between the reaction rate and saturation.

Equation (2) shows that the reaction rate ($r$) depends on the saturation of the solution ($\Omega$) and the reaction surface area (S) of the mineral under consideration. However, the kinetic parameters ($k$ and possibly $\theta$ and $\eta$) depend on the physicochemical conditions under which the reaction occurs, such as the pH, temperature, and/or concentration of a given component (i.e., catalytic or inhibitory effects).

The saturation is expressed as follows:

$$\Omega = \frac{IAP}{K} \tag{3}$$

where IPA is the ion activity product of the specified mineral component dissolved. $K$ is the solubility product.

The relationship between the reaction's free energy ($\Delta G_r$ J mol$^{-1}$) and saturation is as follows:

$$\Delta G_r = -RT \, ln\left( \frac{IAP}{K} \right) \tag{4}$$

It can be seen from Equation (2) that the precipitation rate increases as mineral saturation increases. When the reaction is far from equilibrium, the dissolution rate is independent of the reaction's free energy; whereas, when it is near equilibrium, the dissolution rate decreases. At thermodynamic equilibrium, $\Delta G_r$ is equal to zero and the reaction rate is zero.

The relationship between the mineral dissolution or precipitation rate and temperature can be described by Arrhenius law:

$$k_T = A \, \exp\left( \frac{-E_a}{RT} \right) \tag{5}$$

where $E_a$ (J) is the activation energy of the reaction. $A$ $\left( \text{m}^{-2} \text{ s}^{-1} \right)$ is the reaction rate constant at 25 °C. R is the ideal gas constant (8.314 J mol$^{-1}$ K$^{-1}$). $T$ (K) is the reaction temperature.

The reaction rate calculated above is only applicable in pure water (neutral pH). The dissolution and reaction rates of the minerals usually require the presence of H$^+$ (acidic mechanism) and OH$^-$ (alkaline mechanism) ions. For most minerals, the rate constant K contains the above three mechanisms.

$$k = k_{25}^{nu} \exp\left[ \frac{-E_a^{nu}}{R} \left( \frac{1}{T} - \frac{1}{298.15} \right) \right] + k_{25}^{H} \exp\left[ \frac{-E_a^{H}}{R} \left( \frac{1}{T} - \frac{1}{298.15} \right) \right] a_H^{nH} + k_{25}^{OH} \exp\left[ \frac{-E_a^{OH}}{R} \left( \frac{1}{T} - \frac{1}{298.15} \right) \right] a_{OH}^{nOH} \tag{6}$$

where *Nu*, *h* and *oh* represent the neutral, acidic and basic mechanisms respectively. *a* is the ionic activity. *n* is a power exponent (constant).

The rate constant k can also be related to other ions, such as $CO_3^{2-}$ and $HCO_3^-$, which is often seen in the effect of components on carbonate minerals. Therefore, the reaction rate constant under the influence of ions can be written in a unified form:

$$k = k_{25}^{nu} \exp\left[\frac{-E_a^{nu}}{R}\left(\frac{1}{T} - \frac{1}{298.15}\right)\right] + \sum_i k_{25}^i \exp\left[\frac{-E_a^i}{R}\left(\frac{1}{T} - \frac{1}{298.15}\right)\right] \prod_j a_{ij}^{nij} \quad (7)$$

The superscript *i* denotes an additional mechanism, and *j* is the ion that plays a major role in this mechanism (it can be a basic ion or another type of ion).

### 3.3.2. Mineral Reaction Surface Area

The dissolution of the mineral surface is not uniform, and the dissolution rates of the different parts of the mineral surfaces are different [40,41]. Only a part of the surface of each mineral can be dissolved, and the rest is non-reactive. The mineral surface directly involved in the dissolution or precipitation process is called the reaction surface area or the effective surface area [42], which is often much smaller than the total surface area of the mineral. Although atomic force microscopy (AFM) can be used to study the surface reactions from the nanometer to micrometer scales, this fine expression makes the calculation particularly difficult. In most cases, the reaction surface area is measured via nitrogen or krypton adsorption (Brunauer–Emmett–Teller, BET method) [43]. In addition, the reaction surface area can also be calculated using geometric techniques, but the resulting reaction surface area may differ by several orders of magnitude [44]. As is shown in Figure 5, as the main constituent mineral of the Gaoyuzhuang reservoir rocks, the reaction surface areas used in the different methods differ by more than two orders of magnitude. Clay minerals have a larger reaction surface area, and the reaction surface areas used in the different methods differ by more than three orders of magnitude. Therefore, in most studies, the reaction surface area used is mostly determined according to the understanding and knowledge of the scholars [45].

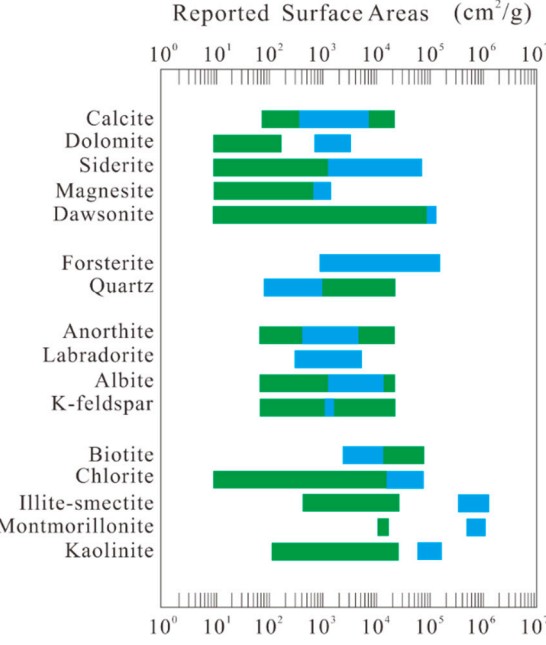

**Figure 5.** Reaction surface area of the different minerals [44]. Blue represents the reaction surface area measured via the BET method and green represents the reaction surface area used in the different reaction transport simulations.

The reaction surface area used in the simulation in this study was based on previous research results. The reaction surface areas of the dolomite, feldspar and other minerals were set as 9.8 cm$^2$/g, and the reaction surface area of the clay minerals was set as 151.63 cm$^2$/g.

### 3.3.3. Thermodynamics Database

It can be seen from Equation (2) that the mineral reaction rate is also affected by the balance. Many scholars have studied thermodynamic data for minerals and have obtained a relatively unified understanding. The thermodynamic database used in this study was created by Blanc [46]. The database was compiled by Thermobridge in December 2020 and contains 85 major ion components, 945 complex ion components, 737 mineral components and 23 gas components.

The main ion components considered in the model were $H^+, Ca^{2+}, Mg^{2+}, Na^+, K^+,$ $Fe^{2+}, H_4SiO_4, HCO_3^-, SO_4^{2-}, Al^{3+}$ and $Cl^-$. The complex ions included all of their complex products. The mineral components included dolomite, quartz, calcite, illite, albite and potassium feldspar (microcline). The reaction formula corresponding to each mineral is as follows:

$$Dolomite(CaMgCO_3) + 2H^+ \leftrightarrow Ca^{2+} + Mg^{2+} + 2HCO_3^- \tag{8}$$

$$Calcite(CaCO_3) + H^+ \leftrightarrow HCO_3^- + Ca^{2+} \tag{9}$$

$$Quartz(SiO_2) + 2H_2O \leftrightarrow H_4SiO_4 \tag{10}$$

$$Illite(Al_{2.35}Mg_{0.25}Fe_{0.25}K_{0.85}H_2O_{12}) + 8.4H^+ + 1.6H_2O$$
$$\leftrightarrow 2.35Al^{3+} + 0.85K^+ + 0.25Fe^{2+} + 0.25Mg^{2+} + 3.4H_4SiO_4 \tag{11}$$

$$Albite(AlNaSi_3O_8) + 4H^+ + 4H_2O \leftrightarrow Al^{3+} + Na^+ + 3H_4SiO_4 \tag{12}$$

$$Microcline(AlKSi_3O_8) + 4H_2O + 4H^+ \leftrightarrow Al^{3+} + K^+ + 3H_4SiO_4 \tag{13}$$

The equilibrium constant of the minerals also varied with temperature, and the relationship with temperature was as follows:

$$\log(K) = a \cdot \ln(T) + b + c \cdot T + \frac{d}{T} + \frac{e}{T^2} \tag{14}$$

In Equation (14), $a$, $b$, $c$, $d$ and $e$ are coefficients, and $T$ is the temperature ($K$).

Table 4 presents the coefficients of the minerals. The calculated mineral equilibrium constants from room temperature to 150 °C are shown in Figure 6.

**Table 4.** Calculation coefficients of main mineral equilibrium constants.

| Coefficients | Dolomite | Calcite | Quartz | Illite | Albite | Microcline |
|---|---|---|---|---|---|---|
| $a$ | $2.83 \times 10^2$ | $1.34 \times 10^2$ | $5.39 \times 10^1$ | $3.81 \times 10^2$ | $2.55 \times 10^2$ | $2.46 \times 10^2$ |
| $b$ | $-1.79 \times 10^3$ | $-8.50 \times 10^2$ | $-3.54 \times 10^2$ | $-2.48 \times 10^3$ | $-1.66 \times 10^3$ | $-1.60 \times 10^3$ |
| $c$ | $-2.90 \times 10^{-1}$ | $-1.39 \times 10^{-1}$ | $-4.19 \times 10^{-2}$ | $-3.44 \times 10^{-1}$ | $-2.20 \times 10^{-1}$ | $-2.13 \times 10^{-1}$ |
| $d$ | $9.96 \times 10^4$ | $4.69 \times 10^4$ | $2.18 \times 10^4$ | $1.55 \times 10^5$ | $1.04 \times 10^5$ | $9.92 \times 10^4$ |
| $e$ | $-5.60 \times 10^6$ | $-2.66 \times 10^6$ | $-1.59 \times 10^6$ | $-9.06 \times 10^6$ | $-6.44 \times 10^6$ | $-6.29 \times 10^6$ |

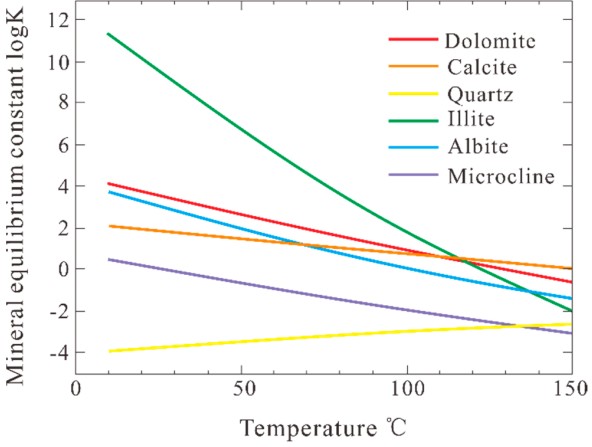

**Figure 6.** Variations in the dissolution equilibrium constants of the main minerals with temperature.

### 3.3.4. Model Building

The experiment was an isovolumetric reaction experiment, and the model was designed to include only a single grid. The volume of the reactor vessel was 500 mL, and the grid was set to the same volume. Since the volume of each sample was only 10 mL, which is negligible relative to that of the reaction vessel, it can be assumed that the reaction conditions were closed, so the model boundary was set as a zero-flow boundary. The temperature settings were consistent with the experimental conditions (40 °C, 60 °C, 80 °C and 100 °C), and the pressure was set as the actual formation pressure (30 MPa). The temperature boundary condition was a zero-flux boundary, which was used to simulate the experimental condition of constant temperature.

According to the quality and density of the actual rock sample and the 25-gram rock sample used in the experiment, the porosity was calculated to be about 98%. The initial mineral composition and content were input according to the actual XRD results, mainly including dolomite, quartz, calcite, albite, microcline and illite. The initial chemical composition of the water, i.e., $H^+$ and $Cl^-$ contents, was calculated ac based on a 15 wt.% HCl solution, which is about 4.11 M, and other ionic components were set to 0.0 M. The simulation time was consistent with the reaction time (1 h). This simulation was a static water-rock interaction, and the simulation tool was TOUGHREACT EOS1 [41].

The initial kinetic parameters of the reaction of each mineral were set according to previous research results [47–49]. As is shown in Equation (6), each mineral corresponded to a different reaction rate constant ($k_{25}$) and activation energy ($E_a$) under neutral and acidic conditions. Table 5 presents the rate constants of the main minerals under the neutral reaction mechanism. The rate constant under the acidic reaction mechanism was obtained by fitting the experimental results. It should be noted that no obvious increase in the $K^+$ and $Na^+$ ion concentrations was observed during the experiment. It is believed that the microcline (potassium feldspar) and plagioclase did not dissolve significantly during the experiment. Therefore, only the reaction rate constants of dolomite, calcite and illite under acidic conditions were corrected.

**Table 5.** Kinetic parameters of the neutral reaction mechanism of the main minerals.

| Minerals | Neutral Reaction Mechanism | |
|---|---|---|
| | Reaction Rate Constant $k_{25}$ mol/m$^2$/s | Reaction Activation Energy $E_a$ kJ/mol |
| Dolomite | $1.1 \times 10^{-8}$ | 31 |
| Quartz | $6.4 \times 10^{-14}$ | 77 |
| Calcite | $1.6 \times 10^{-6}$ | 24 |
| Illite | $3.3 \times 10^{-17}$ | 35 |
| Albite | $5.1 \times 10^{-20}$ | 57 |
| Microcline | $1.0 \times 10^{-14}$ | 31 |

## 4. Results and Discussion

### 4.1. Rock Debris Dissolution Ratio

Table 6 presents the corrosion ratio results for the rock debris for different acid concentrations (15 wt.% and 20 wt.%). Under 60 °C, after 60 min of reaction, the dissolution ratio for 15 wt.% HCl for the three samples was 78.1%, 87.3% and 86.9%, with an average of 84.1%. The dissolution ratios for 20 wt.% HCl were 82.3%, 88.5%, and 87.2%, with an average of 86%.

The difference in the dissolution ratio of the rock cuttings under the same acid concentration is speculated to have been caused by differences in the mineral compositions of the samples. However, in general, the dissolution ratios of the rock debris were greater than 80%, indicating that HCl can produce a good acidizing effect on the Gaoyuzhuang reservoir. The 20 wt.% HCl had a higher dissolution ratio than the 15 wt.% HCl, but their average

difference was only 2.1%. Considering the effect of the acidification and the amount of acid solution, 15 wt.% HCl was finally determined to be the main acid solution.

**Table 6.** Experimental results of the rock debris dissolution for different acid concentrations.

| Sample Depth | Reaction Temperature (°C) | Reaction Time min | 15 wt.% HCl Dissolution Ratio (%) | 20 wt.% HCl Dissolution Ratio (%) |
|---|---|---|---|---|
| 3158–3160 m | 60 | 60 | 78.1 | 82.3 |
| 3174–3176 m | 60 | 60 | 87.3 | 88.5 |
| 3178–3180 m | 60 | 60 | 86.9 | 87.2 |

*4.2. Reactor Experiments*

4.2.1. Mineral Dissolution and Ion Concentration Changes

The main ion concentration of the acid solution after the reaction of core R1 is shown in Figure 7. After the core sample was added to the reactor, the minerals reacted with the acid solution rapidly to produce $CO_2$. The concentrations of $Ca^{2+}$, $Mg^{2+}$, $H_4SiO_4$ in the solution increased rapidly as the reaction progressed. The $Ca^{2+}$ concentration finally stabilized at about 8000 mg/L, $Mg^{2+}$, and $H_4SiO_4$ concentration finally stabilized at about 5000 mg/L. The concentration of $Al^{3+}$ was relatively low and stabilized at about 1000 mg/L.

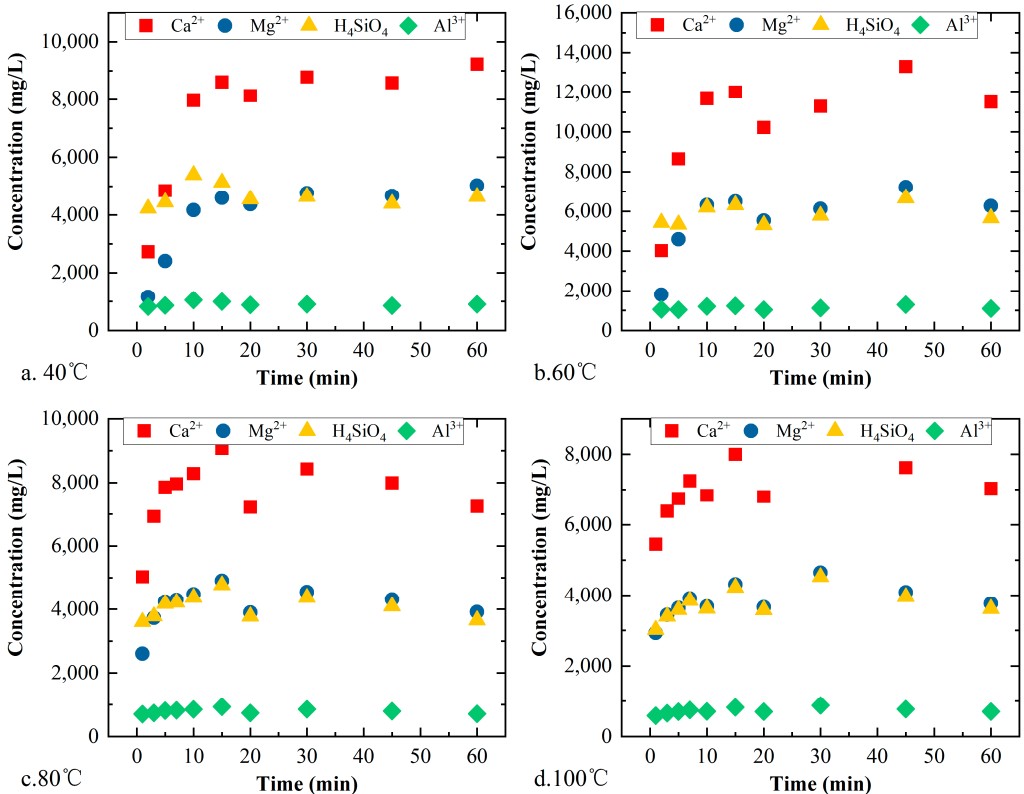

**Figure 7.** Ion concentration of acid after the reaction of core R1 at (**a**) 40 °C; (**b**) 60 °C; (**c**) 80 °C; (**d**) 100 °C.

Based on the mineral composition of the R1 sample, dolomite was the main mineral dissolved, and its main composition was $CaMg(CO_3)_2$. The relative molecular weight of Ca was 40 g/mol, and the relative molecular weight of Mg was 24 g/mol, with a ratio of 1.6. From the change in the acid ion concentration, the trends of the $Ca^{2+}$ and $Mg^{2+}$ concentrations were consistent with that of dolomite dissolution. This indicates that the $Ca^{2+}$ and $Mg^{2+}$ in the acid solution mainly came from the dissolution of dolomite. Calcite

was also one of the sources of $Ca^{2+}$, but the calcite content was low, and its contribution was small. The $H_4SiO_4$ ion content also increased obviously as the reaction progressed, which is speculated to have been caused by the dissolution of clay minerals. The main clay mineral was illite, and the dissolution of 1 mol of illite can produce 3.4 mol of $H_4SiO_4$ and 2.35 mol of $Al^{3+}$, and 0.25 mol of $Mg^{2+}$. The source of the $Al^{3+}$ was also the dissolution of illite.

The main ion concentrations of the acid solution after the reaction of core R2 are shown in Figure 8. As the reaction progressed, the concentrations of $Ca^{2+}$ and $Mg^{2+}$ in the solution increased rapidly. The final content of each ion differed under different conditions; for example, $Ca^{2+}$ finally stabilized at 12,000 mg/L under 40 °C (Figure 8a), at 14,000 mg/L under 60 °C (Figure 8b), at 10,000 mg/L under 80 °C (Figure 8c), and at 9000 mg/L under 100 °C (Figure 8d). This was due to the difference in the rock quality of the initial reaction. However, the ratio of $Ca^{2+}$ to $Mg^{2+}$ after the reaction was stable at about 1.6, indicating that these ions originated from the dissolution of dolomite. Compared with R1, the $Al^{3+}$ and $H_4SiO_4$ contents of R2 were very low, indicating that the clay mineral content of the rock sample was very low. It can also be seen from the XRD test results for R2 that its dolomite content was 99% of the mineral, which is consistent with the experimental results.

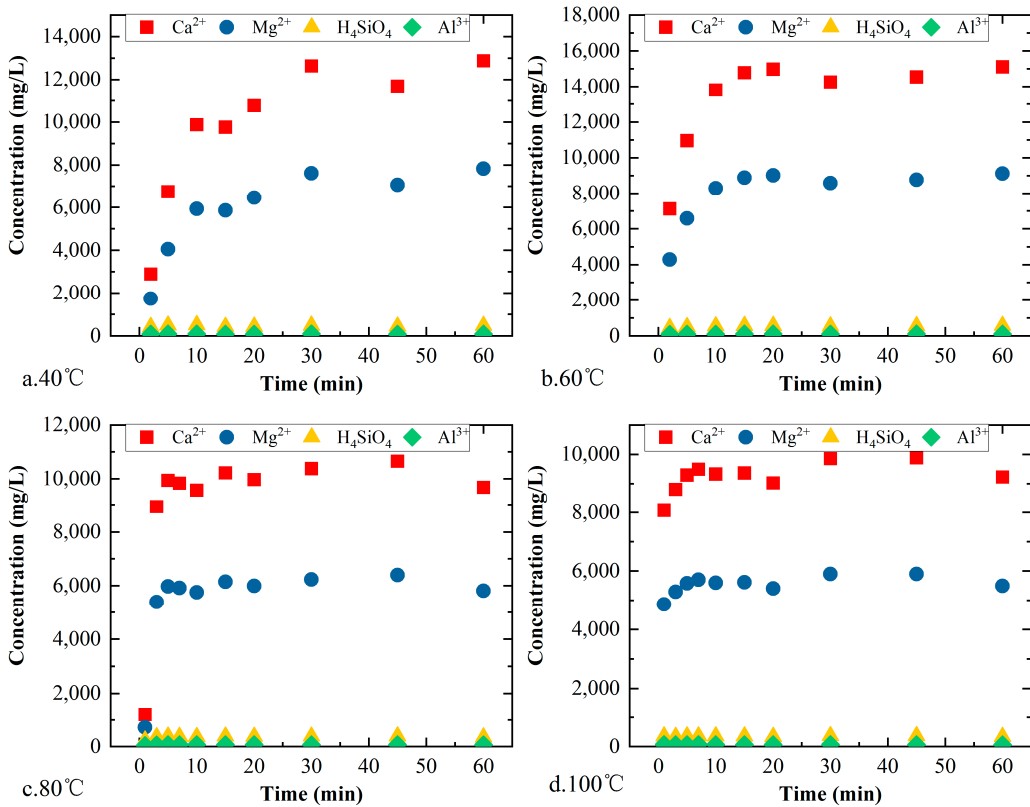

**Figure 8.** Ion concentrations of the acid after the reaction of core R2 at (**a**) 40 °C; (**b**) 60 °C; (**c**) 80 °C; (**d**) 100 °C.

The main ion concentrations of the acid solution after the reaction of core R3 are shown in Figure 9. Similar to the reaction of R1 and R2, $Ca^{2+}$ and $Mg^{2+}$ were the main ions in the solution, and their concentrations reached 10,000 mg/L and 6000 mg/L, respectively. The mineral composition of R3 was mainly dolomite (80%) and quartz (14%). The reaction rate of the quartz was very slow, and the main mineral dissolved in the experiment was dolomite. The clay minerals (5%) (mainly illite) made a small contribution to the $Ca^{2+}$ and $Mg^{2+}$ concentrations. The $H_4SiO_4$ and $Al^{3+}$ were mainly from the dissolution of the clay minerals. The relationship between the $Ca^{2+}$ and $Mg^{2+}$ concentrations revealed an obvious dolomite dissolution process.

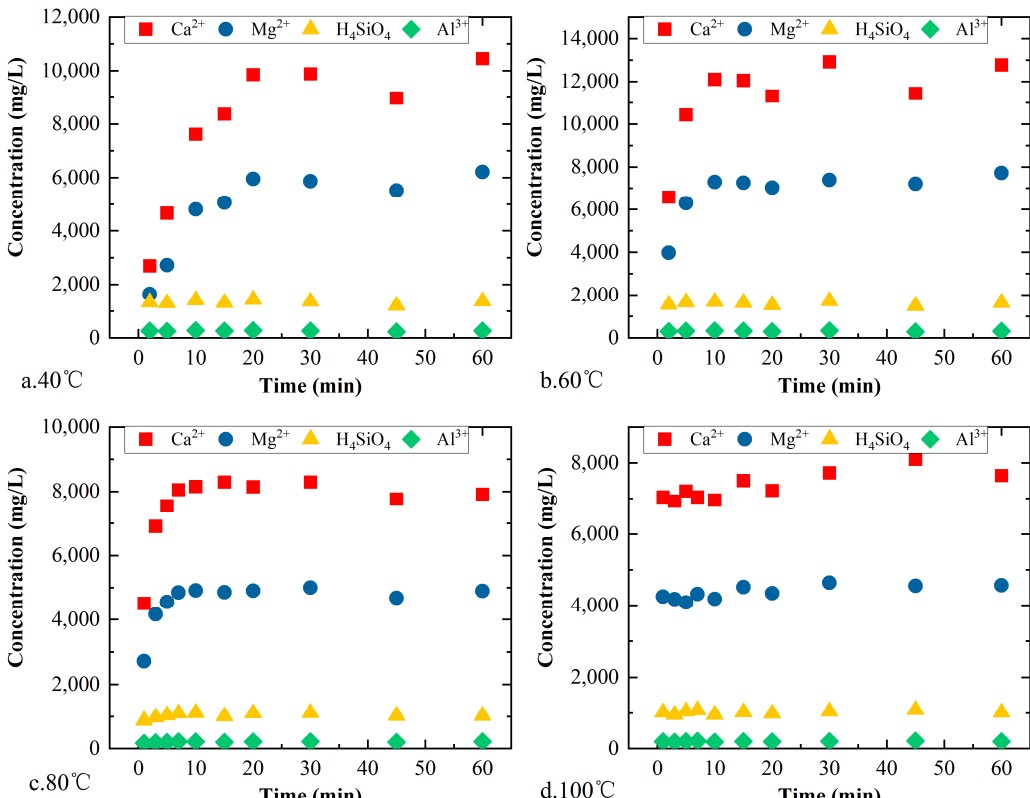

**Figure 9.** Ion concentrations in acid after the reaction of core R3 reaction at (**a**) 40 °C; (**b**) 60 °C; (**c**) 80 °C; (**d**) 100 °C.

In general, the mineral composition of core D22 was relatively simple, mainly including dolomite, quartz and clay. Dolomite was the main mineral dissolved, and its dissolution process determined the change in the $Ca^{2+}$ and $Mg^{2+}$ concentrations. The clay minerals were mainly illite, and their dissolution mainly produced $H_4SiO_4$ and $Al^{3+}$, as well as a small amount of $Mg^{2+}$. When the dolomite content was very high (for example, the dolomite content of R2 was 99%), the ions in the solution were almost completely $Ca^{2+}$ and $Mg^{2+}$, and the $H_4SiO_4$ and $Al^{3+}$ contents were very low.

### 4.2.2. Effect of Temperature on the Dissolution Rate

For core R1, the temperature has a large influence on the mineral reactions. The higher the temperature was, the faster the reaction rate was. Under 40 °C, the core sample was completely dissolved after about 20 min (Figure 7a), and the content of each ion component tended to be stable and did not change. The contents were stable after about 15 min at 60 °C (Figure 7b). For 80 °C and 100 °C, the stabilization times were shorter, only an estimated 10 min and 5 min, respectively (Figure 7c,d).

Temperature also played a significant role in controlling the reaction rate of R2. At 40 °C, all ions reached equilibrium after an estimated 30 min (Figure 8a). They reached equilibrium after an estimated 15 min at 60 °C (Figure 8b), and they reached equilibrium after 10 min and 5 min at 80 °C and 100 °C, respectively (Figure 8c,d). This is consistent with the experimental results for R1.

The effect of temperature on the reaction rate of R3 was very obvious. The stabilization time at 40 °C was an estimated 20 min (Figure 9a), the stabilization time at 60 °C was an estimated 15 min (Figure 9b), and the stabilization times at 80 °C and 100 °C were 10 min and 5 min, respectively (Figure 9c,d).

The temperature had a great influence on the reaction rates of the three minerals. As the temperature increased, the reaction rate increased rapidly. When the temperature reached 100 °C, the rock was completely dissolved in only 5 min.

### 4.3. Mineral Reaction Kinetic Parameters

By continuously adjusting the reaction rate constant of the dolomite under acidic conditions, the simulated and experimental $Ca^{2+}$ and $Mg^{2+}$ contents for core R2 were compared (Figure 10). In general, the simulated $Ca^{2+}$ and $Mg^{2+}$ concentrations were consistent with the experimental values, and the simulation results exhibited good correspondence with the experimental results. Based on the differences in the mineral composition of the rock samples, and the sampling and testing errors, the errors are acceptable. According to the temperature effect, the dissolution rate of the dolomite increased rapidly with the increase in temperature. The simulation results reflected the influence of the better temperature. Through fitting of the data for core R2, the dissolution rate constant k of the dolomite under acidic conditions was calculated to be $k_{25} = 2.4 \times 10^{-4}$ mol/m$^2$/s. This is consistent with the research results of Gautelier [19] and Marty [48].

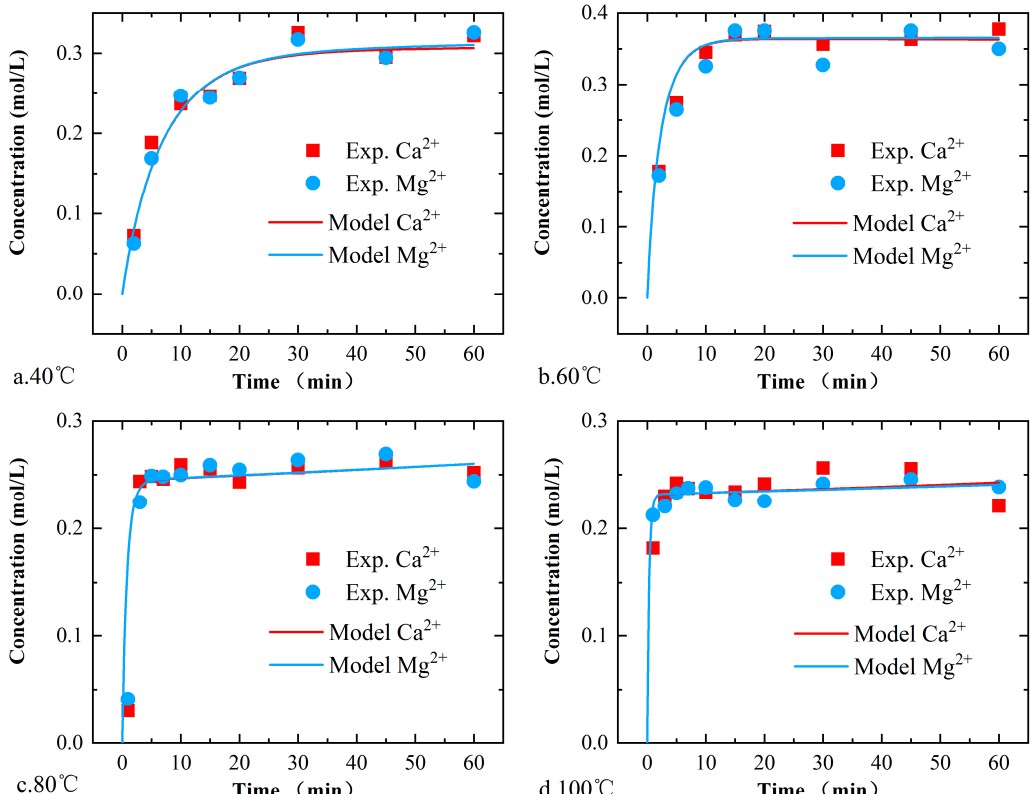

**Figure 10.** Comparison of $Ca^{2+}$ and $Mg^{2+}$ concentrations for core R2.

The simulated changes in the $Ca^{2+}$ and $Mg^{2+}$ concentrations for core R1 are shown in Figure 11. The mineral composition of core R1 was more complex than that of core R2. In addition to dolomite, there were other soluble minerals such as calcite and clay minerals. Therefore, the changes in the $Ca^{2+}$ and $Mg^{2+}$ concentrations were similar but somewhat different. Based on the comparison of the simulated and experimental values, the reaction rate constants of the calcite and illite under acidic conditions match the changes in the $Ca^{2+}$, $Mg^{2+}$ concentrations well.

The $H_4SiO_4$ and $Al^{3+}$ concentrations for core R1 are shown in Figure 12. The above two ions mainly came from the dissolution of illite. Based on the change process, the dissolution rate of illite was much faster than that of dolomite. Finally, the reaction rate constant of illite under acidic conditions matched the $H_4SiO_4$ and $Al^{3+}$ concentrations well.

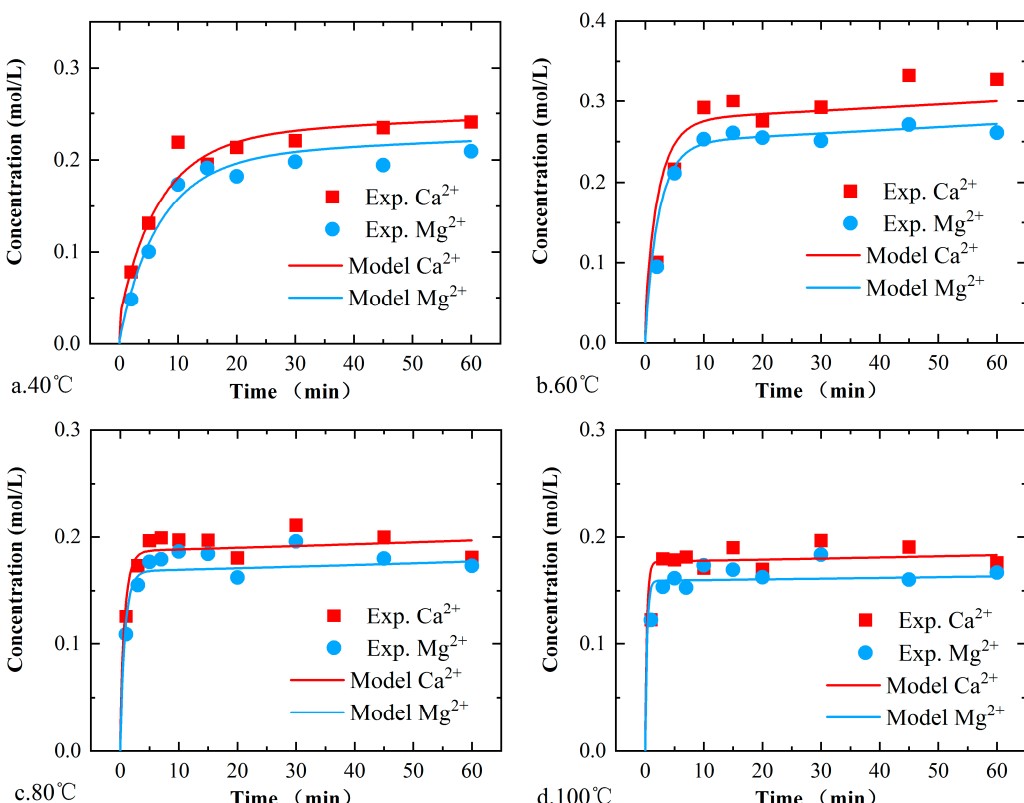

**Figure 11.** Comparison of $Ca^{2+}$ and $Mg^{2+}$ concentrations for R1 core.

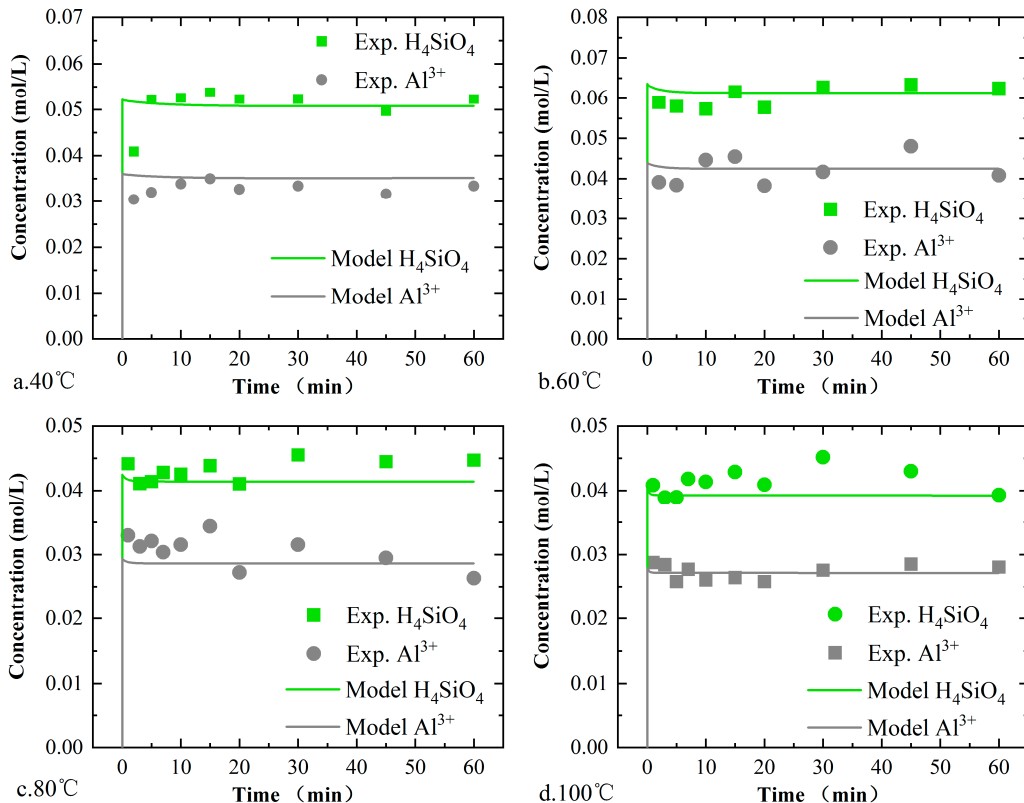

**Figure 12.** Comparison of $H_4SiO_4$ and $Al^{3+}$ concentrations for R1 core.

The kinetic reaction parameters of dolomite, calcite and illite under acidic conditions are shown in Table 7. Under acidic conditions, the reaction rate constant of dolomite

increased by four orders of magnitude, and those of calcite and illite increased by five orders of magnitude and 15 orders of magnitude, respectively. It can be seen that the different reaction mechanisms had a great influence on the dissolution rate of the minerals.

**Table 7.** Kinetic parameters of the acid reaction mechanism for the main mineral components of the rock.

| Minerals | Acid Reaction Mechanism | |
|---|---|---|
| | Reaction Rate Constant $k_{25}$ mol/m$^2$/s | Reaction Activation Energy $E_a$ kJ/mol |
| Dolomite | $2.4 \times 10^{-4}$ | 46 |
| Calcite | $5.3 \times 10^{-1}$ | 14 |
| Illite | $9.5 \times 10^{-2}$ | 36 |

## 5. Conclusions

In this study, the mineral dissolution process of the rocks of Gaoyuzhuang Formation in the Xiong'an New Area during acidizing was investigated, and the kinetic reaction parameters of the main minerals were obtained, providing the basic parameters for the design and evaluation of on-site acidizing. First, three types of core samples from different depths and with different mineral contents were selected, and in situ high-temperature, high-pressure acid–rock reaction experiments were conducted. The reaction experiments were carried out at 40 °C, 60 °C, 80 °C and 100 °C to clarify the ion concentration changes under different experimental conditions and rock mineral compositions. Second, based on transition state theory, an acid-rock reaction kinetic model for the main minerals was established. By fitting the model with experimental results, the kinetic reaction parameters of the main minerals were obtained. The main conclusions of this study are as follows.

(1) The main lithology of the Gaoyuzhuang Formation in the Xiong'an New Area is dolomite, that is, dolomite is the main constituent mineral, followed by quartz and clay minerals.

(2) Hydrochloric acid can produce a good dissolution effect on the Gaoyuzhuang Formation. The average dissolution ratio of 15 wt.% HCl on the rock debris reached 84.1%, so 15 wt.% HCl can be used as the main acid solution for acidizing.

(3) Under the action of hydrochloric acid, the dolomite, calcite and illite were dissolved, generating a large amount of $Ca^{2+}$, $Mg^{2+}$, $Al^{3+}$ and $H_4SiO_4$. The dissolution of the potassium feldspar and plagioclase was not obvious.

(4) The temperature had an obvious effect on the dissolution rates of the minerals. As the temperature increased from 40 °C to 100 °C, the time required for core dissolution to occur decreased from 20 min to 5 min.

(5) The mineral reaction kinetic model based on transition state theory describes the mineral dissolution process well. Under the action of hydrochloric acid (acidic reaction mechanism), the reaction rate constants of dolomite, calcite and illite reached $2.4 \times 10^{-4}$ mol/m$^2$/s, $5.3 \times 10^{-1}$ mol/m$^2$/s and $9.5 \times 10^{-2}$ mol/m$^2$/s, respectively.

**Author Contributions:** Conceptualization and data curation, X.Z.; methodology and visualization, G.Y.; conceptualization and resources, G.W.; resources, F.M.; software, G.Y.; writing—original draft, G.Y.; writing—review & editing, F.M. All authors have read and agreed to the published version of the manuscript.

**Funding:** This research was funded by the Natural Science Foundation of Hebei Province, China (Grant No. D2021504041), National Natural Science Foundation of China (Grant No. 41902310) and China Geological Survey Project (Grant No. DD20221676).

**Acknowledgments:** The authors gratefully acknowledge many important contributions from the researchers of all reports cited in our paper.

**Conflicts of Interest:** The authors declare no conflict of interest.

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
