# Peer review of "Mineral Reaction Kinetics during Aciding of the Gaoyuzhuang Carbonate Geothermal Reservoir in the Xiong’an New Area, Northern China"

_water, doi:10.3390/w14193160_

Round 1

Reviewer 1 Report

^The manuscript deals with deep geothermal energy in China.

In the introduction, deep geothermal energy is generalized as "clean" energy. This is not correct in every case, because deep geothermal energy can release gases like CO2 and H2S.

Not only the minerals in the subsurface determine the permeability of the aquifer and the temperature transition, but also the type of fracturing, fracture opening widths and interconnection are decisive parameters. Nothing is said in the MS about natural karstification.

In maps (Fig 1) it is common to indicate north arrow, coordinates and a scale.

The temperature in wells is given as 60°C at 3000m. Is that 3000 m below ground surface? Then the value of 60°C is surprisingly low.

The authors use the term "kettle". This is unusual, better would be "rection vessel" or "reaction chamber". In this case, it would be even better to use the term "autoclave", because the device can obviously apply not only temperatures up to 100 °C but also pressures up to 60 MPa. This brings us to a critical point of the entire MS: Neither in the title, abstract, introduction nor in the summary is the pressure mentioned. This is also true for the figures (Fig. 7 to 12) where only the temperatures are mentioned but not the pressure. Thus, it remains completely unclear whether the tests were performed at 30 MPa (??) or at ambient condition (0.1 MPa). Thus it is also unclear whether the results are valid for the pressures in the aquifer. In this context, there is also no information about the material the autoclave is made of and how the experiments and sampling under pressure were performed.

Results of thermodynamic modeling are not presented, please add.

Author Response

We greatly appreciate the comments from the reviewer. These comments are of essential benefit in improving the quality of this paper. Our response to you are as follows:

1.In the introduction, deep geothermal energy is generalized as "clean" energy. This is not correct in every case, because deep geothermal energy can release gases like CO2 and H2S.

reply:The original text was indeed not accurate and we changed it to “Geothermal energy will be one of the most important energy resources in the future.”

2.Not only the minerals in the subsurface determine the permeability of the aquifer and the temperature transition, but also the type of fracturing, fracture opening widths and interconnection are decisive parameters. Nothing is said in the MS about natural karstification.

reply:We added the effect of the above factors on reservoir permeability. The statement in the paper was rewritten as “Due to the influence of the regional tectonics and natural karstification, the fracture dis-tribution, type, opening width and interconnection in the Gaoyuzhuang Formation are not homogeneous, leading to non-homogeneous permeability”.

3.In maps (Fig 1) it is common to indicate north arrow, coordinates and a scale.

reply: We added north arrow, coordinates and a scale to Figure 1.

4.The temperature in wells is given as 60°C at 3000m. Is that 3000 m below ground surface? Then the value of 60°C is surprisingly low.

reply: The temperature in wells is given as 60°C at 3000m below ground surface. Because of the homogeneous temperature of the geothermal reservoir due to groundwater, the temperature at a depth of 3000m is not very high. Of course, there are wells in the Xiong’an New Area with temperatures greater than 100 degrees. We have taken it into account in our experiments and simulations.

5.The authors use the term "kettle". This is unusual, better would be "rection vessel" or "reaction chamber". In this case, it would be even better to use the term "autoclave", because the device can obviously apply not only temperatures up to 100 °C but also pressures up to 60 MPa. This brings us to a critical point of the entire MS: Neither in the title, abstract, introduction nor in the summary is the pressure mentioned. This is also true for the figures (Fig. 7 to 12) where only the temperatures are mentioned but not the pressure. Thus, it remains completely unclear whether the tests were performed at 30 MPa (??) or at ambient condition (0.1 MPa). Thus it is also unclear whether the results are valid for the pressures in the aquifer. In this context, there is also no information about the material the autoclave is made of and how the experiments and sampling under pressure were performed.

reply: The word "kettle" is inaccurate and has been corrected in the paper. Our experiments were conducted at 30 MPa pressure and the conditions of the numerical simulations and experiments were the same to ensure that the results match the actual reservoir conditions. We re-specified the pressure conditions in the Abstract and Experimental Methods sections. We added a thermodynamic reaction model for minerals.

6.Results of thermodynamic modeling are not presented, please add.

reply: The thermodynamic model was included in the reaction kinetic model. It was a part of the reaction kinetics model. Our model was built based on Blanc's thermodynamic database. We added thermodynamic reaction models for major minerals.See Equation 7-12.Figure 6 shows the variation of the equilibrium constants of the major minerals with temperature, also as a result of the thermodynamic model.

In addition, we have made a thorough touch-up of the language of the article.

Reviewer 2 Report

This paper focuses on the reconstruction of the geothermal reservoir of the Gaoyuzhuang Formation in Xiongan New Area. The authors investigated the acid rock reaction kinetics of dolomite using experiments and numerical simulations. The reaction kinetic parameters of the main minerals were obtained. The results provide support for the further development and utilization of geothermal resources in Xiongan New Area.

In general, the research significance of the thesis is clear, innovative, with reasonable research methods and correct conclusions. The comments are as follows.

1. It is suggested to modify the language, some expressions are not clear.

2. How the rock samples for experiments were selected in 3.1.

3. How were the experimental temperature conditions determined in section 3.2.

4. In section 3.3.2, why the measured mineral reaction surface area was not used.

5. The article only carries out the experiments of the reactor, why the flow reaction experiment is not carried out?

6. Have the experiments and simulations taken into account the effect of CO2 gas, and how is it considered?

Author Response

We greatly appreciate the comments from the reviewer. These comments are of essential benefit in improving the quality of this paper. Our response to you are as follows:

1.It is suggested to modify the language, some expressions are not clear.

reply:We have made a thorough touch-up of the language of the article.

2.How the rock samples for experiments were selected in 3.1.

reply:Due to the limited deep core sampling, we selected three different mineral types and representative rock samples for the experiment. There are significant differences in the mineral composition of each type. This will make the experimental results more comprehensive.

  1. How were the experimental temperature conditions determined in section 3.2.

reply:The temperature in well D22 and D16 is 60°C at 3000m below ground surface. The temperature of geothermal wells in the Xiong‘’an New Area is different. Some geothermal wells have a temperature greater than 100 degrees. The temperature range chosen for our experiments was representative of the temperature of the geothermal reservoir.

  1. In section 3.3.2, why the measured mineral reaction surface area was not used.

reply:The reaction surface areas used in the different methods differ by more than two orders of magnitude. Clay minerals have a larger reaction surface area, and the reaction surface areas used in the different methods differ by more than three orders of magnitude. So, the reaction surface area used is mostly determined according to the understanding and knowledge of thescholars. We refered to the previous studies that the reaction surface areas of the dolomite, feldspar and other minerals were set as 9.8 , and the reaction surface area of the clay minerals was set as 151.63 .

  1. The article only carries out the experiments of the reactor, why the flow reaction experiment is not carried out?

reply:We conducted comprehensive high-temperature and high-pressure reactor experiments. Flow reaction experiments can measure the dissolution of rocks under acid flow conditions. A combination of these two experimental methods is best. However, due to time and effort constraints, we did not perform flow reaction experiments. Moreover, according to previous studies, the results of these two experiments have a high similarity. We will consider adding flow reaction experiments to future studies to make the reaction kinetic parameters more accurate.

  1. Have the experiments and simulations taken into account the effect of CO2 gas, and how is it considered?

reply:We have taken into account the effect of CO2 gas. Both calcite and dolomite reacted to generate bicarbonate. Carbonic acid equilibrium exists in aqueous solutions. This equilibrium is influenced by the partial pressure of co2.

Round 2

Reviewer 1 Report

some spelling errors (marked in yellow)

Author Response

We thank the reviewers for their careful review and we have corrected the errors in the manuscript.